# Structure and function of the Zika virus full-length NS5 protein

Baoyu Zhao[1,*], Guanghui Yi[2,*], Fenglei Du[1,*], Yin-Chih Chuang[2], Robert C. Vaughan[2], Banumathi Sankaran[3], C. Cheng Kao[2] & Pingwei Li[1]

The recent outbreak of Zika virus (ZIKV) has infected over 1 million people in over 30 countries. ZIKV replicates its RNA genome using virally encoded replication proteins. Nonstructural protein 5 (NS5) contains a methyltransferase for RNA capping and a polymerase for viral RNA synthesis. Here we report the crystal structures of full-length NS5 and its polymerase domain at 3.0 Å resolution. The NS5 structure has striking similarities to the NS5 protein of the related Japanese encephalitis virus. The methyltransferase contains in-line pockets for substrate binding and the active site. Key residues in the polymerase are located in similar positions to those of the initiation complex for the hepatitis C virus polymerase. The polymerase conformation is affected by the methyltransferase, which enables a more efficiently elongation of RNA synthesis *in vitro*. Overall, our results will contribute to future studies on ZIKV infection and the development of inhibitors of ZIKV replication.

[1] Department of Biochemistry and Biophysics, Texas A&M University, College Station, Texas 77843, USA. [2] Department of Molecular and Cellular Biochemistry, Indiana University, Bloomington, Indiana 47405, USA. [3] Molecular Biophysics and Integrated Bioimaging, Berkeley Center for Structural Biology, 1 Cyclotron Road, Lawrence Berkeley National Lab, Berkeley 94720, USA. * These authors contributed equally to this work. Correspondence and requests for materials should be addressed to C.C.K. (email: ckao@indiana.edu) or to P.L. (email: pingwei@tamu.edu).

Zika virus infection has caused human birth defects and Guillain-Barré Syndrome[1,2]. ZIKV belongs in the genus *Flavivirus* of the *Flaviviridae* family, which also includes the important human pathogens Japanese encephalitis virus (JEV) and the Dengues virus (DENV)[3]. The flavivirus genome is a positive-sense RNA of 11-kb in length that contains a 5′ cap structure but lacks a polyA tail. The RNA encodes a long open reading frame that is translated into a polyprotein that is subsequently processed by viral and host proteases into three structural and seven nonstructural proteins[3].

Nonstructural protein 5 (NS5) is essential for the replication of the flaviviral RNA genome[4–6]. The N-terminal portion of NS5 contains a methyltransferase (MT), followed by a short linker that connects to the RNA-dependent RNA polymerase (RdRp). The MT adds the 5′ RNA cap structure to facilitate translation of the polyprotein and to decrease elicitation of the host innate immune response[7–9]. The RdRp initiates RNA synthesis by a *de novo* mechanism wherein a single-nucleotide triphosphate serves as a primer for nucleotide polymerization[10–12]. Herein we report the crystal structure of the Zika virus NS5 protein and the structure of the RdRp domain. The MT was found to affect the conformation of the RdRp domain and increase RNA synthesis.

## Results

**Crystal structure of the ZIKV NS5.** We expressed the full-length NS5 from ZIKV strain MR766 that was originally isolated from Uganda Africa and determined its crystal structure at 3.0 Å resolution (Table 1, Supplementary Fig. 1). The polypeptide chains are well defined except for the N-terminal four residues and the C-terminal 16 residues (Fig. 1a, Supplementary Fig. 2). The MT is complexed with *S*-adenosyl-L-homocysteine (SAH),

and the RdRp adopts a classic 'right-hand' structure consisting of three subdomains: fingers, palm and thumb (Fig. 1a,b). Two zinc ions are found in the fingers subdomain and at the junction of the palm and thumb subdomains of the RdRp.

The overall structure of the ZIKV NS5 has striking similarities to that of the JEV NS5, with the RMS deviation of 0.55 Å for 751 Cα atoms (Fig. 2a). The MT of both proteins are also located at an acute angle to the RdRps. The ZIKV MT is in a distinct orientation relative to the DENV MT. Due to a short $3_{10}$-helix in the linker, the DEN MT is rotated toward the RdRp (Fig. 2b,c). Residues Arg363, Gln598 and Asn576 in the fingers subdomain of the ZIKV RdRp interact with the linker to prevent it from being more flexible (Fig. 2d).

Additional interactions between the MT and the RdRp contribute to the orientation of the MT in the ZIKV NS5. In the MT, residues 112–128 that forms Loop 9, α6 and β4 interacts with the α14, Loop 32 and Loop 40 in the RdRp domain (Supplementary Fig. 3). Notably, residues in these structures are highly similar in the NS5 proteins of the ZIKV and JEV, providing an explanation for the similar orientations of the MT and RdRp in these two proteins (Supplementary Fig. 3a). In addition, Loop 32 of the DENV NS5 was disordered, likely contributing to the altered orientation of the MT in the DENV NS5.

**The NS5 MTase domain.** The MT of ZIKV NS5 adopts a classic α/β/α sandwich structure[13]. The ZIKV MT can be superimposed with the MTs from other flaviviruses with RMS deviations of <0.73 Å (Fig. 3a). The highly conserved structure allows assignment of the residues that function in RNA cap addition in the ZIKV MT. Residues that bind GTP, catalyse methyl transfer and bind the methyl donor *S*-adenosyl-methionine

**Table 1 | Statistics of crystallographic analyses.**

| | ZIKV NS5 RdRp domain* | ZIKV NS5 full-length* |
|---|---|---|
| *Data collection* | | |
| Space group | $P2_1$ | $P2_12_12$ |
| Cell dimensions | | |
| *a, b, c* (Å) | 121.52, 188.71, 192.54 | 136.50, 197.00, 95.28 |
| α, β, γ (°) | 90.0, 91.99, 90.0 | 90.0, 90.0, 90.0 |
| Resolution (Å) | 3.00 (3.05–3.00) | 3.0 (3.09–3.0) |
| $R_{merge}$ | 14.4% (69.8%) | 15.6% (71.5%) |
| $I/\sigma I$ | 6.0 (1.5) | 6.4 (1.6) |
| Completeness (%) | 98.0 (97.9) | 97.6 (98.2) |
| Redundancy | 2.6 (2.6) | 3.0 (3.1) |
| | | |
| *Refinement* | | |
| Resolution (Å) | 3.00 | 3.00 |
| No. reflections | 169314 | 50511 |
| $R_{work}/R_{free}$ | 0.226/0.260 | 0.231/0.268 |
| No. atoms | 40,565 | 14,282 |
| Protein | 40,549 | 14,196 |
| Ligand/ion | 16 | 86 |
| Water | 0 | 0 |
| B-factors | | |
| Protein | 49.5 | 39.5 |
| Ligand/ion | 50.7 | 45.4 |
| Water | NA | NA |
| r.m.s. deviations | | |
| Bond lengths (Å) | 0.002 | 0.002 |
| Bond angles (°) | 0.477 | 0.511 |

NA, not applicable.
*One crystal was used to collect each dataset.
Values in parentheses are for the highest-resolution shells.

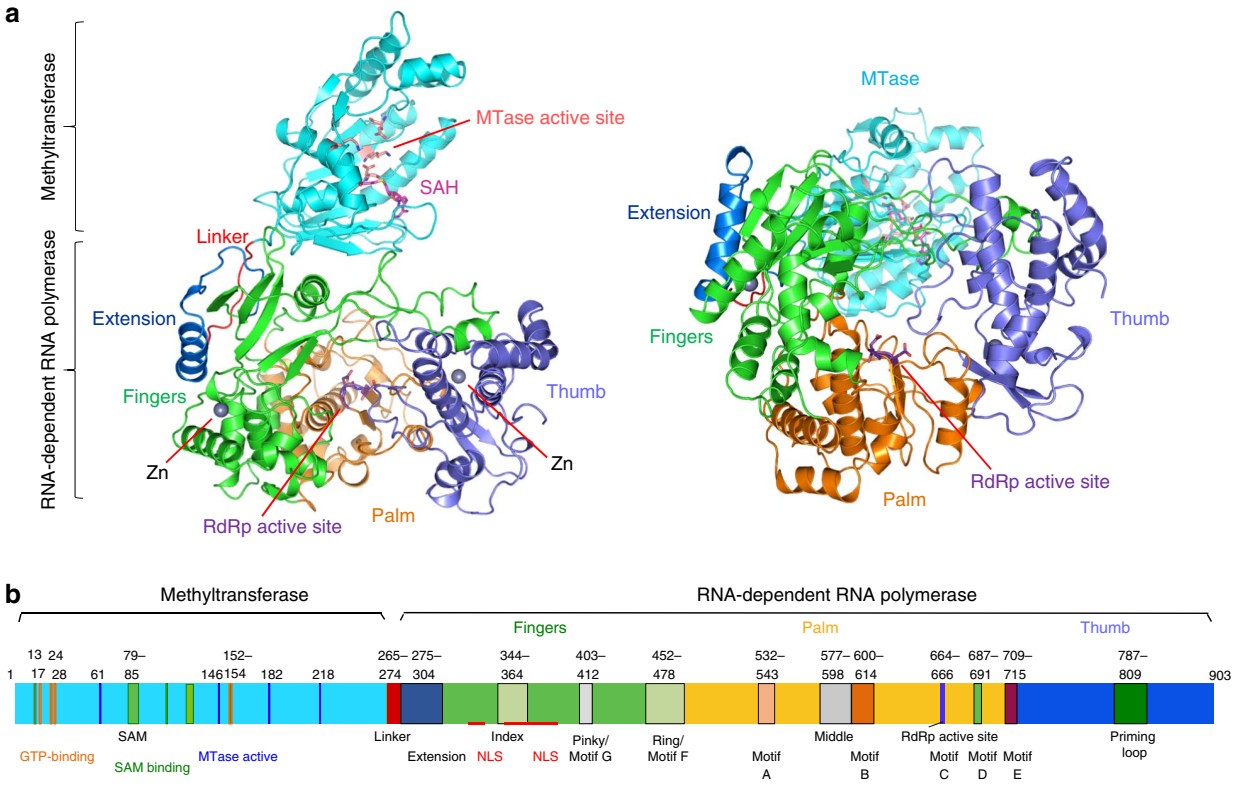

**Figure 1 | Structure of full-length ZIKV NS5.** (**a**) Ribbon representation showing the arrangement of the MT and the RdRp domains of ZIKV NS5. A top view look into the active site of the RdRp is shown on the left and a side view is shown on the right. The MT domain and structural motifs of the RdRp domain are coloured according **b**. The active site residues of the MT and the RdRp are shown by the pink and purple stick representations, respectively. The *S*-adenosyl-L-homocysteine (SAH) molecule that binds to the MT is shown by the magenta stick model. (**b**) Schematic representation of ZIKV NS5 showing the locations of key residues and structural motifs.

(SAM) are arranged in a line within a concave surface of the MT (Fig. 3b,c). The conserved catalytic tetrad of Lys61–Asp146–Lys182–Glu218 that forms the active site is positioned in the centre of the MT (Fig. 3c,d). The putative GTP binding pocket is located to the left of the active site. The SAM-binding pocket is located in a narrow crevice to the right of the catalytic pocket (Fig. 3a,c). SAH, the byproduct of methyl donation from SAM, has the adenylate embedded in a narrow portion of the active site channel, where the Oδ1 of Asp131 forms a H-bond with the N6 of adenine and the Nδ1 of His110 forms a H-bond with the ribose 2′ OH (ref. 7). The homocysteine portion of SAH interacts with loop residues 79–85 (Fig. 3d).

**The NS5 polymerase domain.** Viral RdRps typically have extensive interactions between the fingers and thumb subdomains to encircle the active site of the polymerase[14,15]. Three channels are apparent in the ZIKV polymerase. Based on comparison with polymerases whose ternary structures have been determined and characterized[16], the channels should bind the template RNA (template channel), guide the emergence of the template and nascent RNA (central channel) and enable the entry of the NTPs (NTP channel) (Fig. 4). Motifs A to G that are conserved in sequence and structure, line the active site cavity and contribute to nucleotide and template recognition and nucleotide polymerization.

The active site of the ZIKV RdRp is enclosed due to the interaction between motifs F and G that project from the fingers to contact the thumb in the front of the RdRp

(Fig. 4a,b). The back of the RdRp has a lattice of three loops (Fig. 4a,b). The template RNA will bend at a *ca.* 45° angle and emerge from the central channel. The NTP channel will merge at the confluence of the template and central channels (Fig. 4b,c). Here three aspartates from motifs C and A (Asp535, Asp665 and Asp666) that coordinate divalent metal ions for nucleotide polymerization are localized. The priming loop, which positions nucleotides for polymerization[15], extends from the thumb subdomain is located at the confluence of the three channels (Fig. 4d). In the phage phi6 and HCV RdRp, a tyrosine in the priming element has been shown to form the priming platform by stacking interacting with the initiating nucleotide[16,17]. In the ZIKV NS5, Trp797 likely performs the role of stacking with first nucleotide to facilitate initiation of *de novo* RNA synthesis (Fig. 4d).

The RdRp of the hepatitis C virus (HCV), which belongs to the genus *Hepacivirus* of the *Flaviviridae* family has been extensively studied for the structures required for *de novo* initiation and elongation of RNA synthesis[18]. Residues in the ZIKV RdRp that should contact the RNA and NTPs are located at similar positions to their counterparts in the HCV RdRp ternary complex (Fig. 4e, Supplementary Fig. 4a), suggesting that ZIKV NS5 will have comparable recognition of the template, primer RNA and nucleotides for RNA synthesis. The priming loop of the ZIKV RdRp is larger than that of the HCV RdRp (Supplementary Fig. 4b,c), indicating that conformational changes from the current structure will take place to enable the elongation of the nascent RNA.

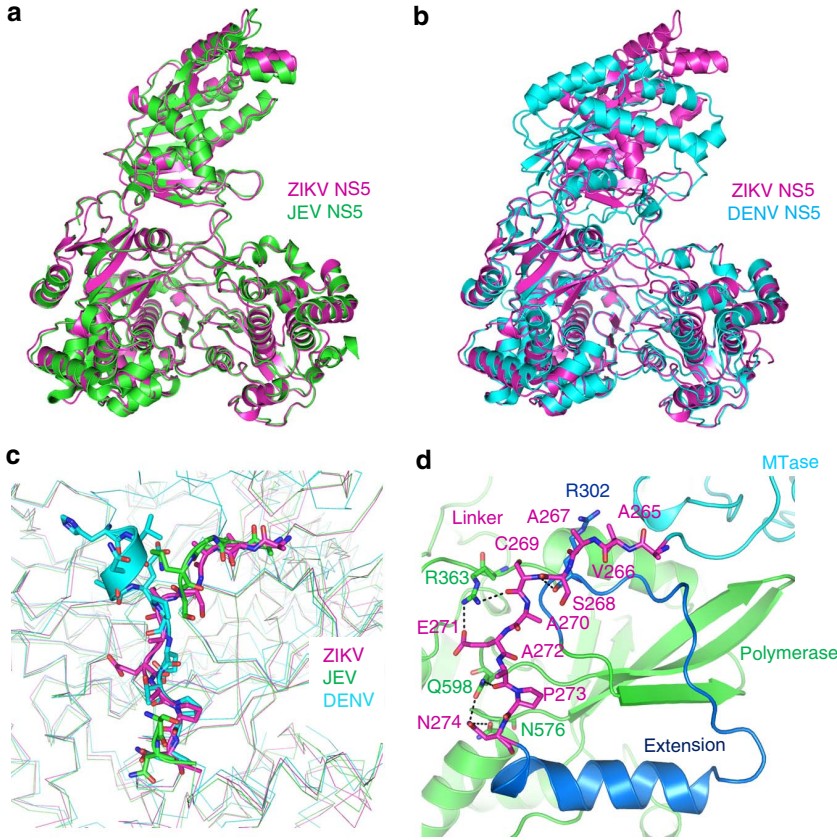

**Figure 2 | Comparison of the NS5 structure of ZIKV to those of JEV and DENV.** (**a**) Superposition of the structures of ZIKV NS5 with JEV NS5 (PDB, 4K6M). (**b**) Superposition of the structures of ZIKV NS5 with DENV NS5 (PDB, 4V0Q). (**c**) Distinct conformations of the linkers in ZIKV, JEV and DENV NS5 that are responsible for the altered orientations of the MT and RdRp domains in these proteins. The linkers are shown as stick models. The backbones of the MTs and RdRps that flank the linkers are shown as thin lines. Note that several residues of the JEV NS5 linker were not resolved. (**d**) Interactions between ZIKV NS5 linker and the fingers subdomain. The linker is shown as stick models (magenta). The extension (blue), portions of the MT (cyan) and the fingers subdomain (green) are shown as ribbons. Key residues R363, Q598 and N576 from the fingers subdomain that interact the linker are shown as sticks. Dashed lines indicate distance of <3.5 Å.

**MTase interacts with the polymerase to affect RNA synthesis.** The MT of the ZIKV NS5 connects to the fingers subdomain of the RdRp and overhangs the NTP channel of the RdRp (Fig. 5a). The MT interacts with the fingers subdomain of the RdRp primarily through a hydrophobic network that involves residues Pro113, Leu115 and Trp121 from the MT and Tyr350, Phe466 and Pro584 from the RdRp (Fig. 5b). The total buried surface area between the MT and the RdRp is ~1,600 Å². The close proximity of the MT to the RdRp suggests that the MT may impact RNA synthesis by the RdRp.

To examine whether the interaction of the MT with the RdRp will affect RNA synthesis, we compared the RNA synthesis activity of NS5 to that of a truncated protein, Δ264, which lacks the MT (Supplementary Fig. 5a). The full-length ZIKV NS5 could initiate RNA synthesis *de novo* or elongate from a primed template in processes that will require distinct RdRp conformations (Fig. 5c, Supplementary Fig. 5b,c). NS5 that had the two aspartates in motif C replaced with alanines was unable to direct RNA synthesis either by *de novo* initiation or by elongation from a primed template (Supplementary Fig. 5b,c). Δ264 synthesized approximately half of the *de novo*-initiated RNA product as did full-length NS5 (Fig. 5c, Supplementary Fig. 6). However, with the template that directs elongative RNA synthesis, Δ264 synthesized sevenfold less product than did NS5. Our result demonstrates that the MT contributes to RNA synthesis by the RdRp, especially for elongative RNA synthesis.

The orientation of the MT relative to the template channel and the central channel suggests that it will affect RdRp interaction with the template RNA. A reversible crosslinking, mass spectrometric assay was used to map residues in NS5 that contact the template RNA[19]. The peptides from NS5 that contacted the RNA were mapped to the template channel, the fingers subdomain and all motifs in the RdRp except for motifs F and D (Fig. 5d, Supplementary Fig. 7a,b). Interestingly, the MT, especially the residues adjacent to the fingers subdomain of the RdRp, also had extensive contact with the template RNA.

**Altered RdRp structure in the absence of MTase domain.** To better understand the difference in RNA synthesis by NS5 and the RdRp, we determined the crystal structure of Δ264 at 3.0 Å resolution. The asymmetric unit contains eight RdRps that exist in two conformations (referred to as conformation 1 and 2) that vary in the locations of motif G, Loop 21 (residues 312–323) and Loop 51 (residues 742–750) that connect the thumb and fingers subdomains (Fig. 5e). The two conformations are affected by residue Arg483 that lies in the template channel. In conformation 1, the side chain of Arg483 inserts between motifs B and F and interacts with the carbonyl backbone of Gly604 and Trp476 (Fig. 5e, Supplementary Fig. 8a). In conformation 2, the locations of motifs B and F prevents the

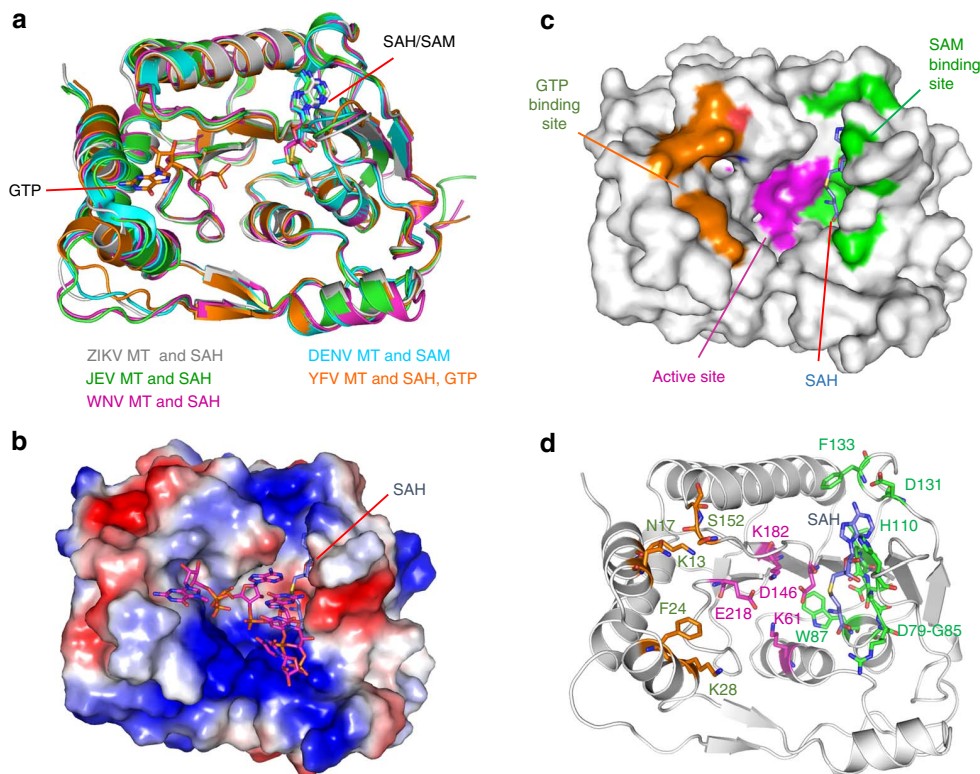

**Figure 3 | Structure of ZIKV NS5 methyltransferase domain (MT).** (**a**) Comparison of the structures of the MT domains of ZIKV, DENV (PDB, 3P97), YFV (PDB, 3EVC), WNV (PDB, 2O Y0) and JEV (PDB, 4K6M). SAH or SAM and GTP bound to the MT domains are shown by the stick models. (**b**) Surface of ZIKV NS5 MT involved in Cap-0 RNA binding coloured by electrostatic potential. Positively charged surface are coloured blue and negatively charged surface red. The Cap-0 RNA $(5'\text{-}^{m7}G_{0ppp}A_1G_2U_3U_4G_5U_6U_7\text{-}3')$ is modelled into the ZIKV NS5 MT by superposition of the DENV MT/Cap-0 RNA complex structure (PDB, 5DTO) onto ZIKV MT. (**c**) Surface representation of ZIKV NS5 MT showing the active site and the binding sites for GTP and SAM. (**d**) Key residues of ZIKV NS5 MT essential for GTP binding (orange), SAM binding (green) and catalysis (magenta). SAH is shown by the blue stick model.

insertion of the side chain of Arg483. Instead, Arg483 interacts with Ala409 of motif G (Fig. 5e, Supplementary Fig. 8b). Full-length NS5 only has one conformation in the corresponding region and it resembles that of conformation 2 (Fig. 5e, Supplementary Fig. 8c). Notably, motif F in NS5 binds with the MT and has a different conformation relative to that in Δ264 (Fig. 5f). The net effect of the presence of the MT is that the RdRp has a reconfigured template channel and possesses a more open NTP channel (Fig. 5f,g). These changes will likely decrease RNA synthesis.

**RNA synthesis by pandemic ZIKV NS5.** The current pandemic ZIKV have been observed to be associated with serious human illness[1]. Strain Brazil/PE243/2015 that was identified from a patient from Recife in Brazil, has more than 35 amino acid substitutions in the NS5 when compared to MR766 (Fig. 6a,b). In terms of RNA synthesis *in vitro*, NS5 proteins from Brazilian/PE243/2015 and MR766 have comparable activities to direct *de novo*-initiated and elongative RNA synthesis (Fig. 6c). Mapping of residues changed in the NS5 protein of the Brazil/PE243/2015 onto the MR766 NS5 structure reveals that the substitutions are located on the surface of NS5 and are not involved in the core of the RdRp that can affect RNA synthesis (Fig. 6d).

**Discussion**
We have determined the crystal structures of the Zika virus NS5 protein and the isolated polymerase domain. The

NS5 protein used was functional for RNA synthesis *in vitro*, producing RNA that initiated *de novo* from the 3′ terminal nucleotide of an exogenously provided template and also elongated from a primed template. The MT domain was shown to bind to the template RNA, and its presence increased elongative RNA synthesis by the RdRp domain. The RdRp alone was also competent for RNA synthesis, but was less active for RNA synthesis *in vitro* (Fig. 5c). This result is in contrast to that of the DENV NS5 protein, where the removal of the MT was shown to increase RNA synthesis *in vitro*[20]. However, the NS5 proteins of the ZIKV and the DENV differ in the network of interactions and the orientations of the MT and RdRp domains. These differences may affect RNA synthesis by the resulting recombinant proteins.

The structure of the ZIKV NS5 protein reveals remarkable similarities with the equivalent structures of other viruses from the *Flaviviridae* family. The active sites for MT activity and RNA synthesis are especially well-conserved. These results suggest that inhibitors of viral MT activity and/or RNA synthesis can be developed to inhibit ZIKV replication. In fact, nucleoside analogues that can inhibit the polymerases of Dengue virus, West Nile virus and yellow fever virus have recently been shown to affect RNA synthesis by ZIKV NS5 (refs 5,21–24). Sofosbuvir, which is highly effective in treating hepatitis C, also has modest inhibitory activity for the replication of the ZIKV[25]. Elucidating additional structures of the ZIKV NS5 complexed to nucleotides and also molecular dynamic studies with the ZIKV NS5 protein with potential inhibitors could aid in the development of inhibitors with higher specificity and potency.

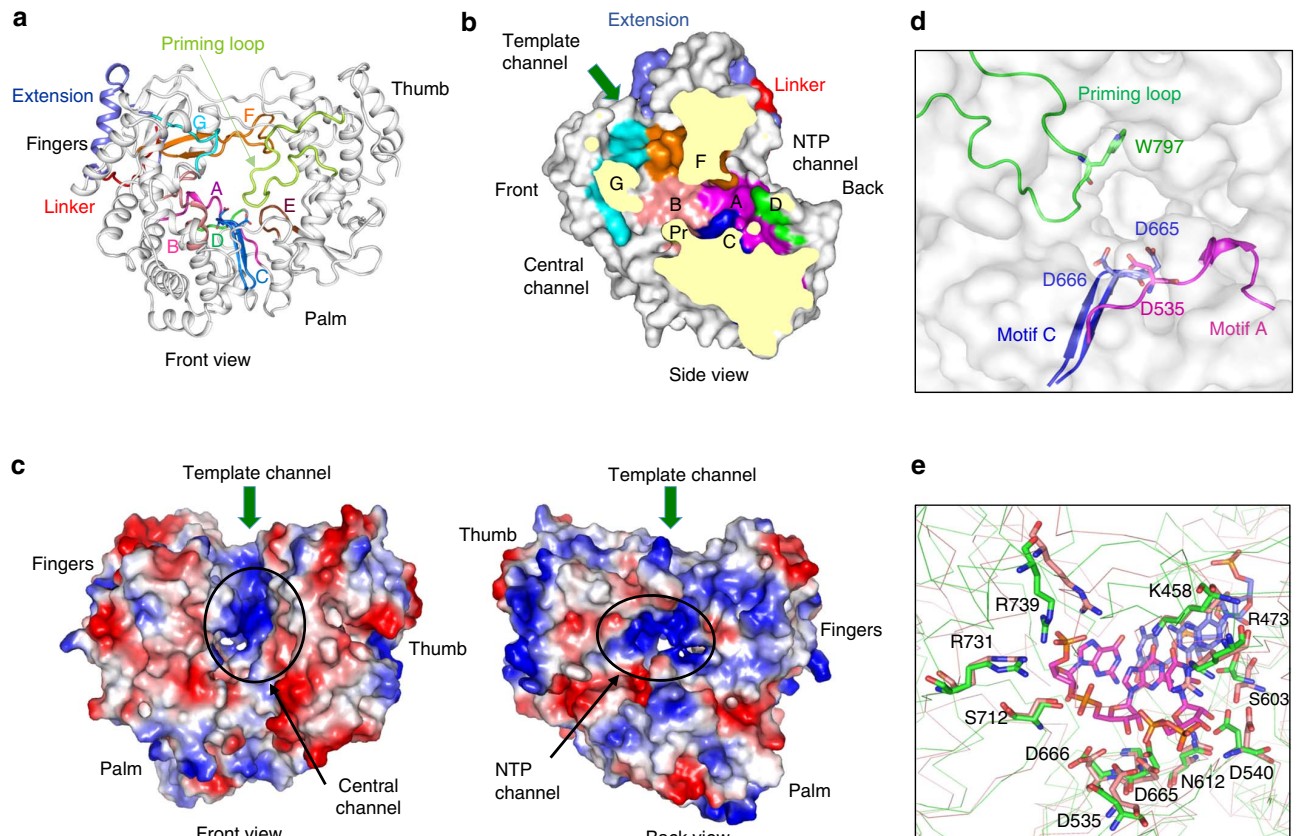

**Figure 4 | Structure of ZIKV RNA-dependent RNA polymerase domain (RdRp).** (**a**) Ribbon representation of the RdRp showing the locations of structural motifs that are critical for RNA synthesis. The extension (slate) is a unique structure of flaviviral NS5 connecting the RdRp with the MT via the linker. The priming loop (lime) extending from the thumb subdomain, forming a platform to coordinate with the NTP for polymerization. (**b**) Cut-away surface representation of ZIKV RdRp showing the locations of the template channel, the central channel and the NTP channel. Motifs G and F that form the encircled active site and also a constriction in the template channel are coloured cyan and orange. The priming loop is identified by 'Pr'. Motif C and A that bind divalent metal ions are coloured blue and purple. (**c**) Electrostatic surface of ZIKV RdRp in two orientations. Positively charged surface is coloured blue and negatively charged surface red. (**d**) Locations of the key residues in the priming loop and active site of ZIKV RdRp. (**e**) Superposition of HCV RdRp (salmon) in complex with template RNA (slate sticks) and the initiation NTP (purple sticks, PDB, 4WTL) with ZIKV RdRp (green). Conserved residues of ZIKV RdRp are shown by the green sticks.

While the Zika virus was initially detected in 1947 in the forest of Uganda Africa, serious illness associated with Zika virus infection was first recorded in Micronesia in 2011 (refs 1,26). At present, the basis for the increased illness associated with the more recent Zika virus outbreak remains to be established. Our comparison of the recombinant NS5 proteins from Africa and from Brazil revealed similar levels of RNA synthesis. In addition, the residues of the Brazilian ZIKV that differ from those of the MR766 virus from Africa are mostly on the surface of the NS5 protein and are less likely to affect the mechanism of RNA-dependent RNA synthesis. The changes, however, could impact interactions with other ZIKV proteins or with cellular proteins.

## Methods

**Chemicals.** SAH was from Sigma and dissolved in $H_2O$. Isopropyl β-D-1-thioga-lactopyranoside, imidazole, dithiothreitol, formaldehyde and all other chemicals are from Sigma. Trypsin was purchased from Promega. Ni-NTA resin was from Invitrogen. $\alpha$-$^{32}$P-CTP and $\alpha$-$^{32}$P-ATP were purchased from PerkinElmer. RNAs were synthesized from Integrated DNA Technologies.

**Gene construction.** DNA encoding NS5 from ZIKV MR766 (GenBank: NC_012532.1) and Brazilian Zika virus PE243/2015 (GenBank: KX197192.1) were chemically synthesized (Integrated DNA Technologies). The cDNA sequences are in Supplementary Table 1. The cDNA fragment was subcloned into

a pET-SUMO vector. The plasmids were transformed into *Escherichia coli* BL21 Rosetta(DE3) pLysS (Novagen) for protein expression. N-terminal truncations of NS5 that lacked the MT (Δ264) were generated via polymerase chain reaction using the forward primer of the sequence of 5′-ACAGAGAACAGATTGGTGGTGCTGT GGCAAGCTGTGCTGAGGGCT-3′ and the reverse primer with the sequence of 5′-CGGATCCGTTATCCACTTTTACAACACTCCGGGTGTGGACCCTTC-3′. Mutations of the RdRp active site were generated via site-directed mutagenesis using the forward primer with the sequence of 5′-CGTATGGCCGTGAGCGG CGCTGCTTGTGTAGTGAAGCCAATTGA-3′ and the reverse primer of the sequence 5′-TCAATTGGCTTCACTACACAAGCAG-CGCCGCTCACGGC CATACG-3′ the QuickChange II kit (Agilent Technologies).

**Recombinant protein production and purification.** Recombinant proteins were expressed in *E. coli* Rosetta (DE3) pLys cells (Novagen) grown in Difco LB broth containing ampicillin (50 μg ml$^{-1}$) and chloramphenicol (17 μg ml$^{-1}$). When the cultures reached an $OD_{600}$ of 1.0–1.3, the temperature was reduced to 16 °C, and isopropyl β-D-1-thiogalactopyranoside was added to a final concentration of 0.4 mM. After a 20 h incubation, the bacteria were harvested by centrifugation.

The *E. coli* cell pellets were suspended in TN buffer (20 mM Tris-Cl pH 8.0, 500 mM NaCl) containing 10 mM imidazole and lysed by sonication. After clarification of the lysate by centrifugation at 15,000g for 30 min, the supernatant was loaded onto a Ni-NTA column that was pre-equilibrated with TN buffer. The column was subsequently washed with TN buffer containing 5 mM β-mercaptoethanol (TNB) and 25 mM imidazole, then TNB buffer containing 40 mM imidazole. The protein was eluted with the TNB buffer containing 350 mM imidazole then exchanged into the TNB buffer containing 20% glycerol. The SUMO-NS5 fusion protein was treated with SUMO protease overnight at 4 °C. NS5 was separated from SUMO and the protease by passage through a second

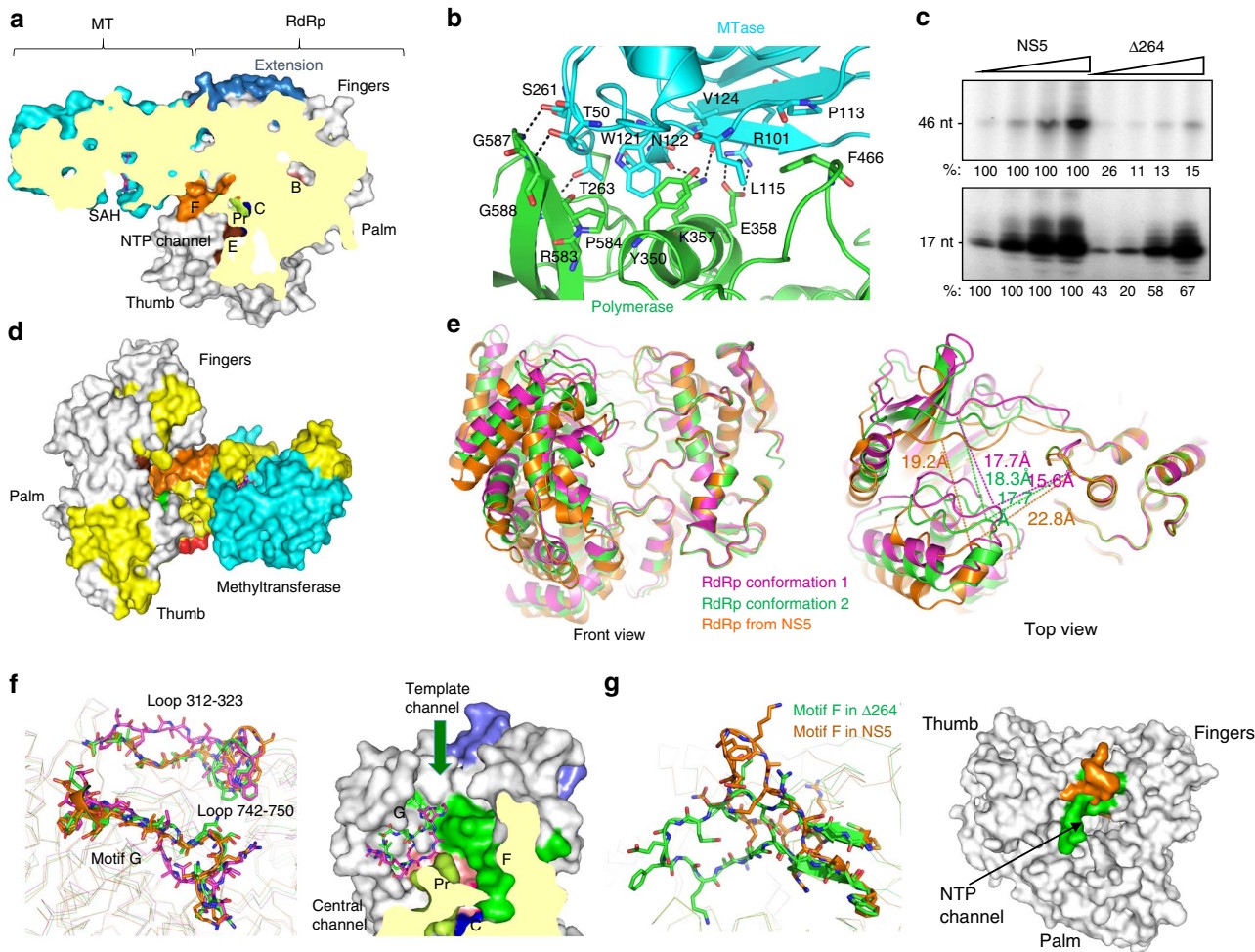

**Figure 5 | The MT affects RNA synthesis by the ZIKV RdRp.** (**a**) Cut-away surface representation showing the locations of the MT and the RdRp in full-length ZIKV NS5. The MT overhangs the NTP channel and contacts the fingers subdomain of the RdRp. (**b**) Interactions between the MT domain (cyan) and the fingers subdomain (green). Dashed lines indicate distance <3.5 Å. (**c**) *In vitro* RNA synthesis catalysed by full-length ZIKV NS5 and Δ264 that lacks the MT. Each set of reactions were performed with 5, 20, 100 and 200 ng of NS5 protein or Δ264 (Supplementary Fig. 6). The PE of 46-nt denotes an elongated product RNA. DN denotes the 17-nt product RNA that initiated *de novo* with a NTP from the 3′-most template nucleotide. The templates used for RNA synthesis are shown in Supplementary Fig. 5. The relative amounts of the products made by Δ264 are normalized to those generated by the same concentration of the enzyme in the reaction with NS5. The results shown are reproducible in four independent assays. (**d**) Regions of ZIKV NS5 that contact the template RNA (PE46) for elongative RNA synthesis. Residues from peptides that are reversibly crosslinked to PE46 are shown in yellow. The structure shown is oriented to show the view at the back of the RdRp that connects to the MT. (**e**) Conformational changes of the RdRp in the absence of the MT. Comparison of eight Δ264 structures in one asymmetric unit reveals two distinct conformations in loop 312–323 and loop 742–750 located in the back of the RdRp and Motif G. Conformation 2 of Δ264 is similar to that in full-length NS5. (**f**) The different conformations of motif F in full-length NS5 (orange) and isolated RdRp (Δ264, green). (**g**) Surface representation showing the different conformations of motif F in full-length NS5 (orange) and isolated RdRp (Δ264, green).

Ni-NTA column. Solutions of NS5 were concentrated and purified with a Superdex200 gel filtration column (GE Healthcare) that was pre-equilibrated with the R buffer (20% glycerol, 20 mM Tris-Cl pH 7.5, 5 mM β-mercaptoethanol) containing 500 mM NaCl. Δ264 that lack the MT was purified using the same protocol except that the Superdex200 column used R buffer containing 150 mM NaCl. All purified proteins were concentrated and stored at −80 °C.

**Crystallization.** ZIKV NS5 at 5 mg ml⁻¹ and with a 10 molar excess of SAH was used for crystallization. Crystals were grown in 0.1 M bis–tris pH 5.5, 1.0 M ammonium sulfate and 1% (wt/vol) PEG 3350. ZIKV NS5 Δ264 containing the RdRp domain was crystallized in 0.1 M Tris-HCl pH 7.5, 0.2 M sodium citrate tribasic dehydrate and 17% PEG 3350. All proteins were crystallized by hanging-drop vapor-diffusion method at 4 °C. The crystals were flash-frozen in liquid nitrogen in the reservoir solution containing 25% glycerol.

**Data collection and structure determination.** Diffraction data were collected at beamline 5.0.2 of the advanced light source with a Pilatus 6M detector. The data were processed with iMOSFLM[27] and merged with Aimless in the CCP4 package[28]. The structure of full-length NS5 was determined by molecular replacement using homology models of NS5 MT and RdRp generated with Swiss-model as search models. JEV NS5 structure (PDB 4k6m) was used to generate the homology model of the MT. An SAH molecule was docked into the difference map of the MT. The structure of Δ264 was determined by molecular replacement using the homology model of NS5 RdRp as the search model. Phaser in the Phenix package was used for structural determination[29]. The models were manually adjusted using Coot[30] and refined using the Phenix package. Statistics from crystallographic analyses of the two structures are in Table 1. All figures depicting structures were generated using PyMOL (http://www.pymol.org).

***In vitro* RNA-dependent RNA polymerase assay.** RNA synthesis assays were performed in 20 µl of reaction containing 20 mM Tris-HCl (pH 7.8), 100 ng of purified ZIKV protein, 1 mM MnCl₂, 5 mM DTT, 0.05% Triton X-100, 10 µM each of ATP, UTP and GTP and 33 nM of α-³²P-CTP. Where present, chemically synthesized RNAs DN17 and PE46 (IDT) were, respectively, at 50 and 100 nM.

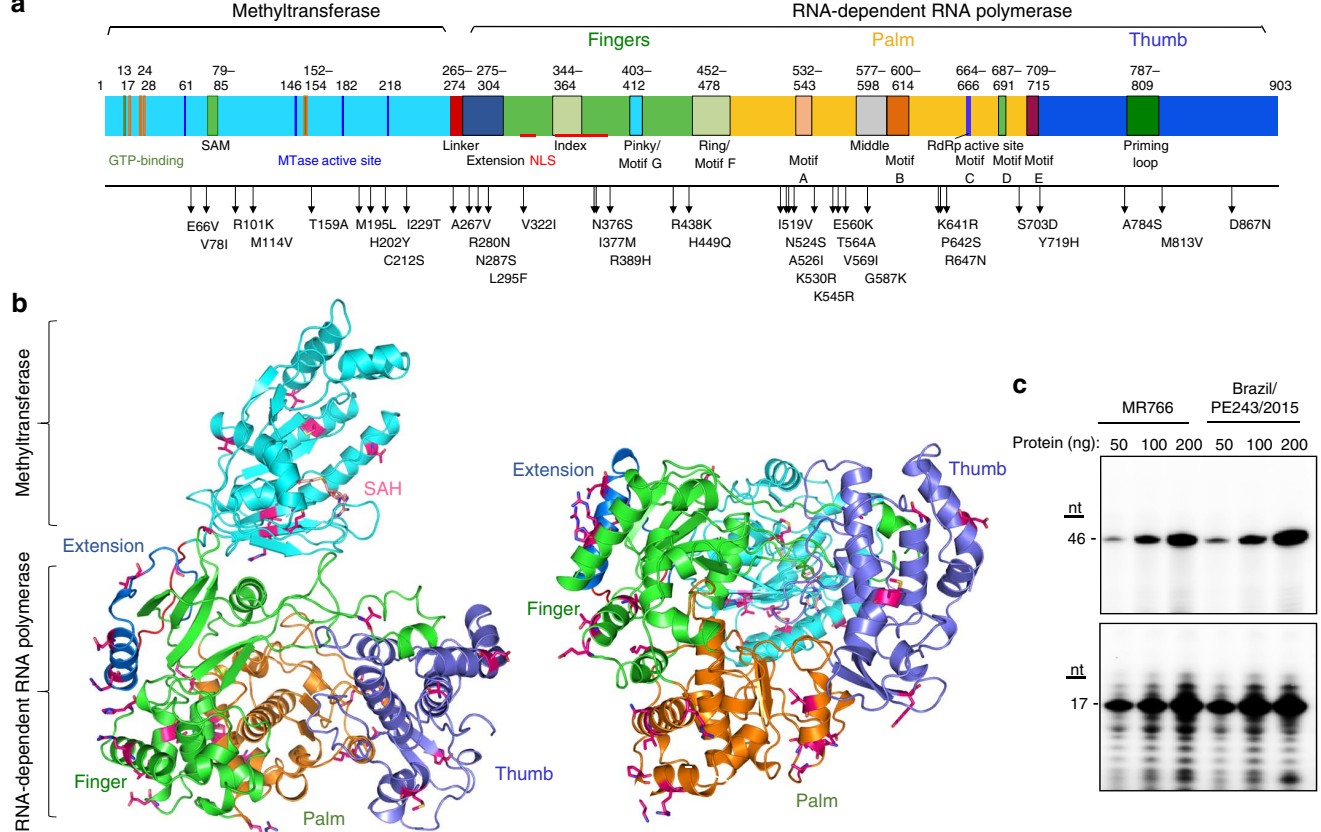

**Figure 6 | NS5 protein from a ZIKV isolated from Brazil has comparable RNA synthesis as the MR766 NS5.** (**a**) Schematic of the ZIKV NS5 showing the locations of the motifs and residues that are different in the NS5 from Brazil/PE243/2015 (GenBank KX197192.1) when compared to the MR766 (NC_012532.1). (**b**) Locations of residues in the Brazil/PE243/2015 in the context of the MR766 NS5 structure. The MT is coloured cyan. The fingers, thumb and palm subdomains in the NS5 RdRp are coloured green, slate and orange, respectively. Residues that differed in the NS5 from Brazil/PE243/2015 are shown in bright red sticks. (**c**) Comparison of RNA synthesis by the NS5 from isolate MR766 and Brazil/PE243/2015. The primer extension (PE) product is of 46-nt. The de novo-initiated (DN) production is of 17-nt. RNAs shorter than full-length are aborted during RNA synthesis or prematurely terminated. The RNAs longer than 17-nt in the DN assay are the result of terminal nucleotide addition to the template RNA. Uncropped gel images are shown in Supplementary Fig. 9.

The reactions were incubated at 30 °C for 90 min, then heated for 3 min at 95 °C and loaded directly onto 22% polyacrylamide gels that contained 7.5 M urea and 1/2× TBE buffer. The gels were separated at 300 V for 3–3.5 h. Radioactive RNA product was detected by a Typhoon Scanner and quantified using ImageQuant software. Radiolabeled RNA markers consisted of chemically synthesized RNAs of 16-, 17- and 18-nt were kinased with γ-$^{32}$P-ATP. Radiolabeled RNA markers of 19- and 46-nt were produced using the recombinant HCV NS5B as described previously[31].

**Mapping NS5–RNA interaction.** Residues in NS5 that contact RNA were mapped using the reversible crosslinking affinity purification assay[19]. ZIKV NS5 2 μM were mixed with RNA PE46 RNA (4 μM) and crosslinked with formaldehyde and processed for mass spectrometry. Control reactions were processed in parallel in the absence of formaldehyde. HPLC–MS analysis was conducted on an LTQ Orbitrap XL mass spectrometer equipped with an Accela HPLC and an electrospray ion source (Thermo Scientific). Peptides were eluted over a 90-min gradient, and tandem MS data was acquired using collision-induced dissociation. Peptides were identified using SearchGUI (v3.1.0)[32], and searched against a concatenated target/decoy database constructed from the cRAP database (http://www.thegpm.org/crap/index.html) of the sequence of ZIKV NS5. Identification settings included an unspecific protease, 10 p.p.m. MS1, and 0.3 Da MS2 error tolerances and oxidation of methionine as a variable modification. MS2 peptide spectrum matches were inferred using PeptideShaker (v1.13.3)[33]. Posterior error probability was calculated in PeptideShaker using the ratio of hits from the decoy database relative to the true database search. Only assignments with high confidence and in two independent replicates and those absent in the control reactions were used.

**Data availability.** The coordinates for the structure of the full-length ZIKV NS5 have been deposited in the Protein Data Bank under the accession code 5U0B.

The coordinates for the structure of the RdRp domain have been deposited in the Protein Data Bank under the accession code 5U0C. The PDB accession codes 4K6M, 4WTL, 3P97, 3ECV, 2OY0 and 5DTO were used in this study. The UniProt accession codes Q32ZE1and ANC90425.2 and the NCBI accession codes NC_012532.1 and KX197192.1 were used in this study. All other data are available from the corresponding authors upon reasonable request.

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

## Acknowledgements

C.C.K. acknowledges seed funding from the Johnson Center for Innovation and Translational Research. We thank Laura Kao for editing the manuscript. The Berkeley Center for Structural Biology is supported in part by the National Institutes of Health, National Institute of General Medical Sciences and the Howard Hughes Medical Institute. The advanced light source is supported by the Director, Office of Science, Office of Basic Energy Sciences, of the U.S. Department of Energy under Contract No. DE-AC02-05CH11231.

## Author contributions

C.C.K., P.L. and B.Z. conceived of the study, designed the experiments, analysed the results, wrote and edited the manuscript. B.Z. and F.D. purified the proteins and generated the crystals for the NS5 and RdRp and solved the structure along with P.L. B.S. collected the X-ray diffraction data. G.Y. made all of the expression constructs, identified the conditions for protein purification and generated the RNA synthesis results. Y.C. developed the initial expression and purification protocol for the Brazilian NS5 protein. R.C.V. performed and analysed the mass spectrometric analysis of peptides in NS5 that contact RNA.

## Additional information

**Competing interests:** The authors declare no competing financial interests.

**Publisher's note**: 

