## [Peer review file · Nature Communications]

Reviewers' comments:

Reviewer #1 (Remarks to the Author):

General comments: Zhao et al. solved the crystal structures of Zika virus FL NS5 and its RdRp domain. They compared these structures with each other, with JEV, and DENV3 FL NS5, as well as HCV RdRp. They demonstrated the elongation and de novo initiation activities of both Zika proteins. They further modeled and mapped the binding sites of the template RNA and nucleoside inhibitor, 2'-F/2'-CH₃-UTP in the Zika FL NS5 and RdRp, respectively. Overall, the conformation of Zika FL NS5 protein is very similar to that of JEV FL NS5. There are however differences, such as the linker region (partially disordered JEV but full resolved in Zika) and orientations of the F motif, loop 312-323, and could be more clearly highlighted (see comments below). The authors should provide some hypothesis or rationale perhaps based on phylogenetic relatedness, as to why the overall conformation of Zika FL NS5 is like JEV instead of DENV3. The weakness in the manuscript is the absence of functional data to interrogate the relevance of the residues involved in inter-domain interaction, RNA and nucleoside diPO₄ analog binding. The writing could be further made improved and made more succinct.

Specific comments:

1. Abstract (page 2, lines 23-24)) and summary (page 7, lines 148-149): The sentence that "Zika NS5 structure bears striking similarity to related Flaviviruses" is inaccurate, as it is similar to JEV, not DENV3. The sentence should be amended and made more specific.
2. Pg 3, line 55: please state if C-alpha how many amino acids were superimposed.
3. Pg 5, line 89: cite reference(s) regarding the roles of W797/H800 in de novo initiation.
4. Pg 5, last paragraph, residues involved in inter-domain interaction: please state of these are the same amino acid observed to form inter-domain interactions in JEV NS5.
5. Pg 6, lines 112-117: ref 22 used a DENV RdRp domain that started from aa273-900, whilst the Zika RdRp protein studied by the authors comprised aa265-903 (inclusive of the inter-domain linker). Based on Lim et al 2011 (JBC), the extended DENV RdRp domain (inclusive of linker) should be more active than the FL NS5 protein. Please explain.
6. Extended method, Mapping RNA binding site by cross-linking: was trypsin used to digest the protein prior to mass spectrometry. Please elaborate.
7. Fig 4C, Extended Fig 7C, and extended method: radiolabeled RNA ladders are mentioned but not shown in PAGE gels.
8. Fig 4C and method: 100 ng of Zika FL NS5 and RdRp proteins were indicated for in vitro assay. PAGE gels indicated a range of protein concentrations were used. Please clarify.
9. Extended Fig 7C: please indicate the unit for the protein concentrations stated above the PAGE gel.
10. Pg 6-7: What is the functional or biological significance of the two conformations of motif G, aa312-323, and 742-750 in Zika RdRp? Is residue R485 conserved in all flaviviruses, and is there any functional data on this residue? Please elaborate.
11. Page 8, structure determination line 173: please provide more details on search models, which Flavivirus NS5 structures were used and if the two domains (MTase, RdRp) were searched separately. Were the 8 RdRp molecules restrained for refinement?
12. Resolutions of both Zika protein structures are not very high, and their refinements warrant further improvements.
13. Zika FL NS5 and RdRp validation reports, entry composition: information for only 1 Zn

ion is mentioned, whilst 2 Zn molecules are mentioned in main text.

14. Zika FL NS5 and RdRp validation reports, too-close contacts: many close contacts in crystal structure are <1.5 angstrom and should be re-worked.

15. Zika FL NS5 and RdRp validation reports: Too many rotamers, should be also reviewed. Stereo chemistry of SAH should also be reviewed.

16. Zika FL NS5 and RdRp validation reports: Ramachandran and side chain outliers need to be improved. Clash score for RdRp structure needs to be improved.

Reviewer #2 (Remarks to the Author):

The authors have determined the structure of the ZIKV RdRp at 3A resolution with good collection and refinement statistics.

They compare their structure with other flavivirus, namely the JEV NS5, the DENV NS5 and report the great similarity between these structures.

An interesting point on this article is a change in the relative position of the MT in respect to the NS5 Pol domain, "due to the short 310-helix in the linker in the DENV NS5 (Extended Data Fig. 1C and 1D). Residues Arg363, Gln598, and Asn576 in the fingers subdomain of the ZIKV RdRp interact with this linker to prevent it from being more flexible (Extended Data Fig. 1D)...."

The work is interesting owing to the general interest on ZIKV, and deserves rapid publication.

Major points:

- The authors have to prove that the MTase change in relative orientation is not a crystallographic artefact due to cristal packing.
- In the absence of detailed "material and methods" as well as the unavailability of RNA synthesis assays (not yet published, ref 21), the RNA synthesis assays cannot be trusted: the authors have used an E. coli strain which expresses the T7 RNA Polymerase, and thus, even a minute contamination of the prep with T7 RNA Pol would give rise to the quite poor RNA synthesis activity shown in Fig. 4C.

Minor point:

The autors state (p7) that the Brazilian isolate is more pathogenic than the African isolate (ref1). While it is not said as such in this reference, it is not clear that the viruses have the same pathogenic profile but pathogenic effects by the African isolate escaped attention or were unnoticed for any reason.

Reviewer 1

General comments: ... Overall, the conformation of Zika FL NS5 protein is very similar to that of JEV FL NS5. There are however differences, such as the linker region (partially disordered in JEV but full resolved in Zika) and orientations of the F motif, loop 312-323, and could be more clearly highlighted (see comments below). The authors should provide some hypothesis or rationale perhaps based on phylogenetic relatedness, as to why the overall conformation of Zika FL NS5 is like JEV instead of DENV3. The weakness in the manuscript is the absence of functional data to interrogate the relevance of the residues involved in inter-domain interaction, RNA and nucleoside diPO₄ analog binding. The writing could be further made improved and made more succinct.

Response: We have performed additional analyses of conformation of the ZIKV NS5 and a comparison to those of the JEV and DENV NS5. As the reviewer suggested, sequences involved in the interdomain interaction of the ZIKV NS5 was more similar to those of the JEV NS5 (Extended Data Fig. 3A). In addition, we now illustrate several secondary structures that interact are found in the ZIKV NS5 JEV NS5 that are not shared by the DENV NS5 (Extended Data Fig. 3B). These interactions help to explain the relative orientations of the MT and RdRps in the ZIKV, JEV and DENV NS5 proteins.

Motif F in the ZIKV and JEV are actually in fairly similar positions in the ZIKV and JEV NS5 proteins. No specific statement on motif F is warranted.

Specific comment 1. Abstract (page 2, lines 23-24)) and summary (page 7, lines 148-149): The sentence that “Zika NS5 structure bears striking similarity to related Flaviviruses” is inaccurate, as it is similar to JEV, not DENV3. The sentence should be amended and made more specific.

Response: we have modified the sentence to only refer to JEV (P. 2, lines 23-24).

Specific comment 2. Pg 3, line 55: please state if C-alpha how many amino acids were superimposed.

Response: We have superimposed the new refined ZIKV NS5 structure on JEV NS5 (PDB 4K6M). There are 751 C α that were superimposed with the RMS deviation of 0.55 Å. This is stated on p. 3, lines 57-58.

Specific comment 3. Pg 5, line 89: cite reference(s) regarding the roles of W797/H800 in de novo initiation.

Response: The role of an aromatic residue in the priming structure for de novo initiation has been explained starting on p. 5, line 100. References 18 and 19 have been added.

Specific comment 4. Pg 5, last paragraph, residues involved in inter-domain interaction: please state of these are the same amino acid observed to form inter-domain interactions in JEV NS5.

Response: Analysis of the interactions between the MT and the RdRp are added on p. 4, lines 64-70. Illustration of the interactions is now added as Extended Data Fig. 3.

Specific comment 5. Pg 6, lines 112-117: ref 22 used a DENV RdRp domain that started from aa273-900, whilst the Zika RdRp protein studied by the authors comprised aa265-903 (inclusive of the inter-domain linker). Based on Lim et al 2011 (JBC), the extended DENV RdRp domain

(inclusive of linker) should be more active than the FL NS5 protein. Please explain.

Response: Lim et al. compared the activities of two DENV NS5 truncations that contain residues 265-900 and 272-900. The two proteins crystallized to different symmetric groups and had different activities for RNA synthesis. The linkers in the DENV and ZIKV NS5 contact the body of the RdRp to different extents and we had stated that the DENV linker forms a short helix that altered the orientation of the MT domain (p. 4, line 60). To address the Reviewers comment, we do now state that, in contrast to the result we observed, the deletion of the DENV RdRp is more active for RNA synthesis in vitro (p. 7, lines 128-129).

Specific comment 6. Extended method, Mapping RNA binding site by cross-linking: was trypsin used to digest the protein prior to mass spectrometry. Please elaborate.

Response: Trypsin was indeed used in the analysis. We note, however, that trypsin can cleave C-terminal to residues other than arginine and lysine. The RCAP method was previously cited as ref. 23. The methodology is added in the Supplemental information.

Specific comment 7. Fig 4C, Extended Fig 7C, and extended method: radiolabeled RNA ladders are mentioned but not shown in PAGE gels.

Response: The RNA ladders are now explained in the legend to Extended Data Fig. 5.

Specific comment 8. Fig 4C and method: 100 ng of ZIKV FL NS5 and RdRp proteins were indicated for in vitro assay. PAGE gels indicated a range of protein concentrations were used. Please clarify.

Response: The assay conditions have been added to the legend in Extended Data Fig. 5 and also added to the Supplemental information.

Specific comment 9. Extended Fig 7C: please indicate the unit for the protein concentrations stated above the PAGE gel.

Response: The amount of protein used in the reactions is now clearly labeled in Fig. 8C.

Specific comment 10. Pg 6-7: What is the functional or biological significance of the two conformations of motif G, aa312-323, and 742-750 in Zika RdRp? Is residue R485 conserved in all flaviviruses, and is there any functional data on this residue? Please elaborate.

Response: We observed that $\Delta 264$ which lacks the MT caused the RdRp to be less active for elongative RNA synthesis (Fig. 4C). The analysis of the structures of the RdRp provided the basis for the reduced RNA synthesis. The two distinct conformations of motif G, and residues 312-323, and 420-750 could decrease RNA synthesis by perturbing the template channel and the NTP channel. This was stated on p. 8, lines 150-153.

Specific comment 11. Page 8, structure determination line 173: please provide more details on search models, which Flavivirus NS5 structures were used and if the two domains (MTase, RdRp) were searched separately. Were the 8 RdRp molecules restrained for refinement?

Response: The structure of Japanese Encephalitis virus NS5 (PDB 4K6M) was used as a template to generate homology models of ZIKV NS5 using Swiss-model (<https://swissmodel.expasy.org>). The RdRp of the homology model was used to solve the

structure of ZIKV RdRp structure by molecular replacement with Phaser in the Phenix package. Homology models of the methyltransferase domain and RdRp domain were used in a two model search to solve the structure of full-length ZIKV NS5. This information have been included in the revised manuscript.

Specific comment 12. Resolutions of both Zika protein structures are not very high, and their refinements warrant further improvements.

Response: Further refinement of the structures have been performed and are presented in our revised models.

Specific comment 13. Zika FL NS5 and RdRp validation reports, entry composition: information for only 1 Zn ion is mentioned, whilst 2 Zn molecules are mentioned in main text

Response: We have checked our PDB files submitted for validation at PDB. Full-length NS5 PDB file has 4 Zn ions (chain ID C and Z), while the RdRp structure has 16 Zn ions (chain ID Z). This corresponds to 2 zinc ions for each protein molecule. We also checked the occupancy of all the zinc ions; they are all with occupancy of 1.0. We checked both validation reports. They listed all the zinc ions in the PDB files.

Specific comment 14. Zika FL NS5 and RdRp validation reports, too-close contacts: many close contacts in crystal structure are <1.5 angstrom and should be re-worked.

Response: We have remodeled the structures and refined them with optimized X-ray and stereochemistry weight. The quality of the models has improved significantly. A comparison of qualities of the models before and after recent remodeling and refinement are listed in the table below:

Models	RdRp Old	RdRp New	Full-length old	Full-length New
R _{cryst} (%)	19.7	22.6	21.6	23.1
R _{free} (%)	25.5	25.9	26.2	26.8
Rmsd bond	0.013	0.002	0.009	0.002
Rmsd angle	1.272	0.475	1.006	0.511
Clash Score	11.6	5.64	8.94	4.45
Ramachandran Favored (%)	92.3	94.4	93.6	94.6
Ramachadran Outlier (%)	1.2	0.4	0.8	0.3
Rotamer outlier (%)	16.5	4.21	13.4	6.4
Overall Score	2.98	2.16	2.75	2.21

Specific comment 15. Zika FL NS5 and RdRp validation reports: Too many rotamers, should be also reviewed. Stereo chemistry of SAH should also be reviewed.

Response: We have remodeled the two structures and refined them with optimized X-ray/stereochemistry weights. The stereochemistry of the structures was improved with much less rotamer outliers and lower clash scores (see table above). The stereochemistry of SAH was carefully inspected, remodeled, and checked against high resolutions SAH structures. The previous flag in the validation report was due to a large deviation of the bound angle (118°) between C-C α -C β of one SAH molecule from the ideal bound angle (109.5°), while the bound

length are all closed to theoretical values. In the newly refined structural model these bond angles are 111.1° for both SAH molecules.

Specific comment 16. Zika FL NS5 and RdRp validation reports: Ramachandran and side chain outliers need to be improved. Clash score for RdRp structure needs to be improved.

Response: The quality of both structural models has improved significantly after remodeling and refinement (see table above). Both Ramachandran outliers and sidechain rotamer outliers were significantly reduced. The clash scores are also much better compared to previous models.

Reviewer 2

Major points:

-The authors have to prove that the MTase change in relative orientation is not a crystallographic artifact due to cristal packing.

Response: The JEV NS5 has the same conformation as the ZIKV NS5 despite the fact that its crystal packing was of the H3 space group while the ZIKV NS5 was of the P2₁2₁2 space group, with completely different unit cell parameters and different packing interactions. Therefore, crystal packing cannot account for the similar conformations of the ZIKV and JEV NS5 proteins. The revised manuscript now includes analysis of the interactions of the MT and RdRps in the JEV, DENV, and ZIKV proteins that provide the basis for the JEV and ZIKV NS5 proteins adapting a similar conformation (Extended Data Fig. 3, p. 4, lines 64-70).

-In the absence of detailed "material and methods" as well as the unavailability of RNA synthesis assays (not yet published, ref 21), the RNA synthesis assays cannot be trusted: the authors have used an E. coli strain which expresses the T7 RNA Polymerase, and thus, even a minute contamination of the prep with T7 RNA Pol would give rise to the quite poor RNA synthesis activity shown in Fig. 4C.

Response: The methods for RNA synthesis assay have now been added to the Supplemental information section of the paper. The sequences of the RNA and demonstration of RNA synthesis are shown in modified Extended Data Fig. 5.

It is highly unlikely that contaminating T7 RNA polymerase is responsible for the RNA synthesis we observed. First, our proteins are extensively purified. Second, an active site mutant of the ZIKV NS5 purified in the same manner as the full-length NS5 failed to give RNA synthesis. These results have been added to Extended Data Fig. 5B and 5C. Third, the templates we have designed to analyze activities of the ZIKV NS5 does not resemble the promoter or preferred sequence recognized by the T7 RNA polymerase.

Minor point 1. The autors state (p7) that the Brazilian isolate is more pathogenic than the African isolate (ref1). While it is not said as such in this reference, it is not clear that the viruses have the same pathogenic profile but pathogenic effects by the African isolate escaped attention or were unnoticed for any reason.

Response: Instead of making a claim that the African ZIKV is less pathogenic, we have modified the text on p. 8, lines 154-155 to state: "The current pandemic ZIKV have been observed to be associated with serious human illness.¹".

REVIEWERS' COMMENTS:

Reviewer #1 (Remarks to the Author):

The authors have sufficiently addressed the concerns raised.

Reviewer #2 (Remarks to the Author):

The authors have adequately answered the referees' questions.
The manuscript is seriously improved and should be published.

Reviewer Comments:

Reviewer #1 (Remarks to the Author):

The authors have sufficiently addressed the concerns raised.

Reviewer #2 (Remarks to the Author):

The authors have adequately answered the referees' questions.
The manuscript is seriously improved and should be published.

Response: We thank the reviewers for their effort in guiding the revision of this manuscript.